# microRNA165 and 166 modulate response of the Arabidopsis root apical meristem to salt stress

Daria Scintu[1,2], Emanuele Scacchi[3,6], Francesca Cazzaniga[1,6], Federico Vinciarelli [1], Mirko De Vivo[1], Margaryta Shtin[1,2], Noemi Svolacchia [1], Gaia Bertolotti[1], Simon Josef Unterholzner [4], Marta Del Bianco[5], Marja Timmermans [3], Riccardo Di Mambro [2], Paola Vittorioso[1], Sabrina Sabatini [1], Paolo Costantino[1] & Raffaele Dello Ioio [1✉]

In plants, developmental plasticity allows for the modulation of organ growth in response to environmental cues. Being in contact with soil, roots are the first organ that responds to various types of soil abiotic stress such as high salt concentration. In the root, developmental plasticity relies on changes in the activity of the apical meristem, the region at the tip of the root where a set of self-renewing undifferentiated stem cells sustain growth. Here, we show that salt stress promotes differentiation of root meristem cells via reducing the dosage of the microRNAs miR165 and 166. By means of genetic, molecular and computational analysis, we show that the levels of miR165 and 166 respond to high salt concentration, and that miR165 and 166-dependent *PHABULOSA* (*PHB*) modulation is central to the response of root growth to this stress. Specifically, we show that salt-dependent reduction of miR165 and 166 causes a rapid increase in *PHB* expression and, hence, production of the root meristem pro-differentiation hormone cytokinin. Our data provide direct evidence for how the miRNA-dependent modulation of transcription factor dosage mediates plastic development in plants.

[1] Dipartimento di Biologia e Biotecnologie Charles Darwin, Università di Roma, Sapienza - via dei Sardi, 70, 00185 Rome, Italy. [2] Department of Biology, University of Pisa, via L. Ghini, 13, 56126 Pisa, Italy. [3] Center for Plant Molecular Biology, University of Tübingen, Auf der Morgenstelle 32, Tübingen 72076, Germany. [4] Faculty of Science and Technology, Free University of Bozen-Bolzano, Piazzale Università, 5, 39100 Bolzano, Italy. [5] Italian Space Agency, Rome, Italy. [6] These authors contributed equally: Emanuele Scacchi, Francesca Cazzaniga. ✉email: raffaele.delloioio@uniroma1.it

In plants, development must be both robust—to ensure appropriate growth—and plastic—to enable the adaptation to external cues[1–5]. Plastic development largely depends on the modulation of gene expression, controlling the concentration of developmental factors, such as hormones, transcription factors (TFs) and signaling molecules[6–9]. A classic example of plant developmental plasticity is the adaptation of plant growth to high salt conditions, a stress that inhibits shoot and root development[10]. Roots are the first organs sensing salt concentration in soil, where high salt reduces meristem activity and root growth[2–4].

It has been suggested that the regulation of several plant hormones and miRNAs mediate the plant response to salt stress[3,4,11–14]. However, the molecular interplays mediating the adaptation of plant roots to salt stress are still vague. Post-embryonic root growth is supported by the activity of the root meristem, a region located at the root tip where self-renewing stem cells divide asymmetrically in the stem cell niche (SCN), originating transit-amplifying daughter cells that divide in the division zone (DZ)[5]. Once these cells reach a developmental boundary denominated transition zone (TZ), they stop dividing and start to elongate in the so-called elongation/differentiation zone[5]. A dynamic balance between cell division and cell differentiation ensures continuous root growth, maintaining a fixed number of cells in the DZ. Alterations in this dynamic equilibrium promote or inhibit root growth[5,15].

microRNA molecules (miRNA) play a key role in the control of plant development, regulating robust and plastic development since embryogenesis[16–18]. Maturation of plant miRNAs depends on the activity of a multiprotein complex (microprocessor complex) comprising the DICER-LIKE1 (DCL1), HYPONASTIC LEAVES1 (HYL1) and SERRATE (SE) proteins that cut pre-miRNA transcripts into 21 nucleotides mature miRNA[14]. In the root miRNAs are master regulators of cell developmental stages. For instance, miR160 targeting AUXIN RESPONSE FACTORS 10, 16, and 17 maintains stem cell function[19], miR396 controlling the expression of the GROWTH REGULATING FACTORS TFs regulates transition from stem cells to transit amplifying cells[20]; miR156 that targets SQUAMOSA PROMOTER LIKE controls root differentiation program[21]; miR393 regulating mRNA stability of TRANSPORT INIHBITOR RESISTANT1/AUXIN F-BOX in response to nutrient availability controls root plastic development[22]. Among the miRNAs that regulate root meristem development, miR165 and 166 have been shown to be main regulator of root development[23,24]. miR165 and miR166 are pleiotropic regulators of plant developmental processes. miR165 and 166 family consists of nine independent loci (MIR165 A-B and MIR166 A-G) that drive expression of pre-miR165 and 166 in different tissues and at different developmental stages[25]. miR165/166 activity is crucial in the control of robust development, restricting the expression of the HOMEODOMAIN LEUCINE ZIPPER III (HD-ZIPIII), including PHABULOSA (PHB) and PHAVOLUTA (PHV), which are involved in root and shoot development, vascular growth, and leaf and embryo polarity[23,24,26,27]. In the root, miR165/166 regulate meristem homeostasis and radial patterning[23,24]; pre-miR165A and pre-miR166A/b transcription is promoted by the SCARECROW (SCR) and SHORTROOT TFs[23] and, thanks to the cell-to-cell mobility, mature miR165 and 166 diffuse to pattern both the root vasculature and the ground tissue[23,25,28–30].

In the root meristem, the miR165-166-PHB module promotes the synthesis of the plant hormone cytokinin, an important player in root developmental plasticity regulating cell differentiation rate of meristematic cells via the activation of the ARABIDOPSIS HISTIDINE KINASE3 (AHK3)/ARABIDOPSIS RESPONSE REGULATOR 1/12 (ARR1/12) pathway[31,32].

Here, we show that, in response to salt stress, miR165 and 166 modulate PHB expression to adjust root meristem activity. Salt exposure results in changes in cytokinin biosynthesis, which further regulates the miR165/166-PHB module. Hence, in addition to the above-described miRNA activity in controlling root robust development, we provide clear evidence that, in response to environmental cues, miRNAs are crucial in the control of root plastic development, modulating the dosage of TFs.

## Results

**miR165/166 mediate root meristem response to salt stress.** The growth of roots of Arabidopsis seedlings exposed to salt slows down after 5 h of exposure to 150 mM NaCl[3] (Supplementary Fig. 1a). We hypothesized that salt stress might inhibit root growth acting on meristem activity. To verify this, we analyzed during time (5, 8, 16 and 24 h) root meristem size in plants treated with 150 mM NaCl. A significant reduction in root meristem size was detected already after 5 h of treatment (Fig. 1a–f). Analysis of stem cell and cell division markers such as QC25 and CYCB1;1:GUS showed that salt exposure does not alter SCN and DZ activity (see Supplementary Fig. 1b–f), suggesting that salt treatment mostly affects cell differentiation activity. To elucidate the molecular mechanisms behind root response to salt stress we first analyzed the role of the miR165/166-PHB module, as this module have been involved in controlling root meristem development in response to external stimuli[24].

MIR165A, MIR166A and B are expressed in the Arabidopsis root meristem and control root meristem activity regulating PHB expression[23–25]. We measured, via quantitative real time PCR (qRT), the mRNA level of PHB and pre-miRNA levels of MIR165A, MIR166A, and MIR166B after short-time treatments (up to 4 h) to 150 mM NaCl. Interestingly, roots subjected to salt treatment showed decreased levels of the pre-miRNA of MIR165A, MIR166A, and MIR166B and higher transcription of PHB mRNA (Fig. 1g–i). Decrease in pre-miRNA occurs already after 30' of salt exposure, preceding the increase of the PHB mRNA, which occurred only after 2 h of salt treatments (Fig. 1g–i; Supplementary Fig. 2d). This time lag suggests that NaCl specifically targets MIR165A, MIR166A and MIR166B expression, which, in turn, regulates PHB mRNA levels.

To substantiate the primary role of miR165/166 in salt stress response, we investigated the response of plants expressing MIR166A under the control of the SCR promoter (SCR::MIR166A) to salt treatment. Since the SCR promoter is insensitive to salt stress and SCR is expressed in the endodermis as MIR166A, B and MIR165A[2,23], SCR-driven MIR166A expression should compensate the decrease of miR166 level caused by NaCl treatment (Supplementary Fig. 2a–c). Indeed, plants carrying the SCR::MIR166A construct displayed longer root meristems than Wt and did not show any reduction in root meristem size even after 5 h of 150 mM NaCl treatment (Fig. 1j–n), suggesting that the downregulation of miR165/166 is required for salt stress response. Concomitantly, the phb phv double loss-of-function mutant, that has a root meristem size resembling the one of SCR::MIR166A plants, showed no reduction in root meristem size after 5 h of NaCl treatments (Supplementary Fig. 2e–i).

Our data suggest that salt stress represses miR165 and 166 levels, causing a subsequent increase of the activity of HD-ZIPIII family members such as PHB. To corroborate this notion, we exposed phb-1d plants to 150 mM NaCl. In these plants, a mutation in the miR165 and 166 target site of the PHB gene prevents the miRNA-dependent PHB repression, and in turn an increase of PHB transcript level, responsible of generating a smaller meristem[33–35]. We reasoned that if salt stress goes

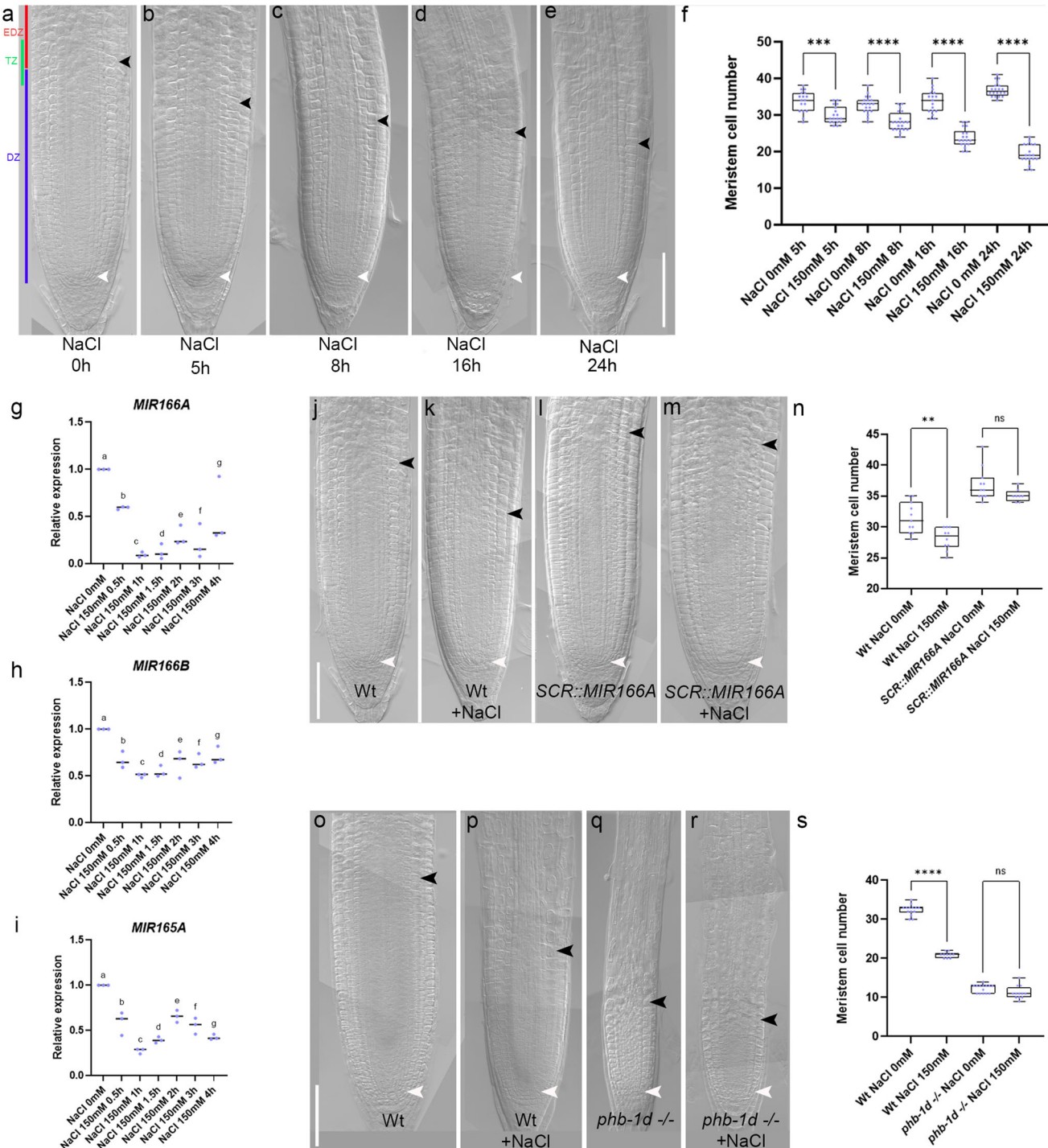

**Fig. 1 Salt stress inhibits root meristem activity regulating miR165 and 166 levels. a–e** DIC of 5 days post germination (dpg) Wt root meristems exposed to 150 mM NaCl for 5, 8, 16, and 24 h. **f** Root meristem cell number of Wt root meristems exposed to 150 mM NaCl for 5, 8, 16, and 24 h (***$p = 0.0001$, ****$p < 0.0001$; One way ANOVA with post hoc Sidak's multiple comparison test; $n = 17, 18, 19, 20, 17, 18, 20, 18$). qRT–PCR analysis of *preMIR166A* (**g**), *preMIR166B* (**h**) and *preMIR165A* (**i**) RNA levels in the root tip of Wt plants upon 150 mM NaCl for 30 m, 1 h, 1.5 h, 2 h, 3 h and 4 h (One way ANOVA with post hoc Dunnett's multiple comparisons test; $n = 3$; different letters show statistical significance). DIC of 5 dpg root meristems of Wt (**j**), Wt exposed to 150 mM NaCl for 5 h (**k**), *SCR::MIR166A* (**l**) and *SCR::MIR166A* exposed to 150 mM NaCl for 5 h (**m**). **n** Root meristem cell number of Wt and *SCR::MIR166A* root meristems exposed to 150 mM NaCl for 5 h (ns not significant, **$p = 0.0021$; One way ANOVA with post hoc Sidak's multiple comparison test; $n = 11, 10, 11, 8$). DIC of 5 dpg root meristems of Wt (**o**), Wt exposed to 150 mM NaCl for 5 h (**p**), *phb-1d −/−* (**q**) and *phb-1d −/−* exposed to 150 mM NaCl for 5 h (**r**). **s** Root meristem cell number of Wt and *phb-1d −/−* root meristems exposed to 150 mM NaCl for 5 h (ns not significant, ****$p < 0.0001$; One way ANOVA with post hoc Sidak's multiple comparison test; $n = 14, 10, 14, 12$). Box and whiskers plots show the median, 25th and 75th percentile (box limits), the 10th and 90th percentiles (whiskers), and outliers points. Scale Bar 100 µm, white arrowheads indicate the cortical stem cell, black arrowheads the TZ.

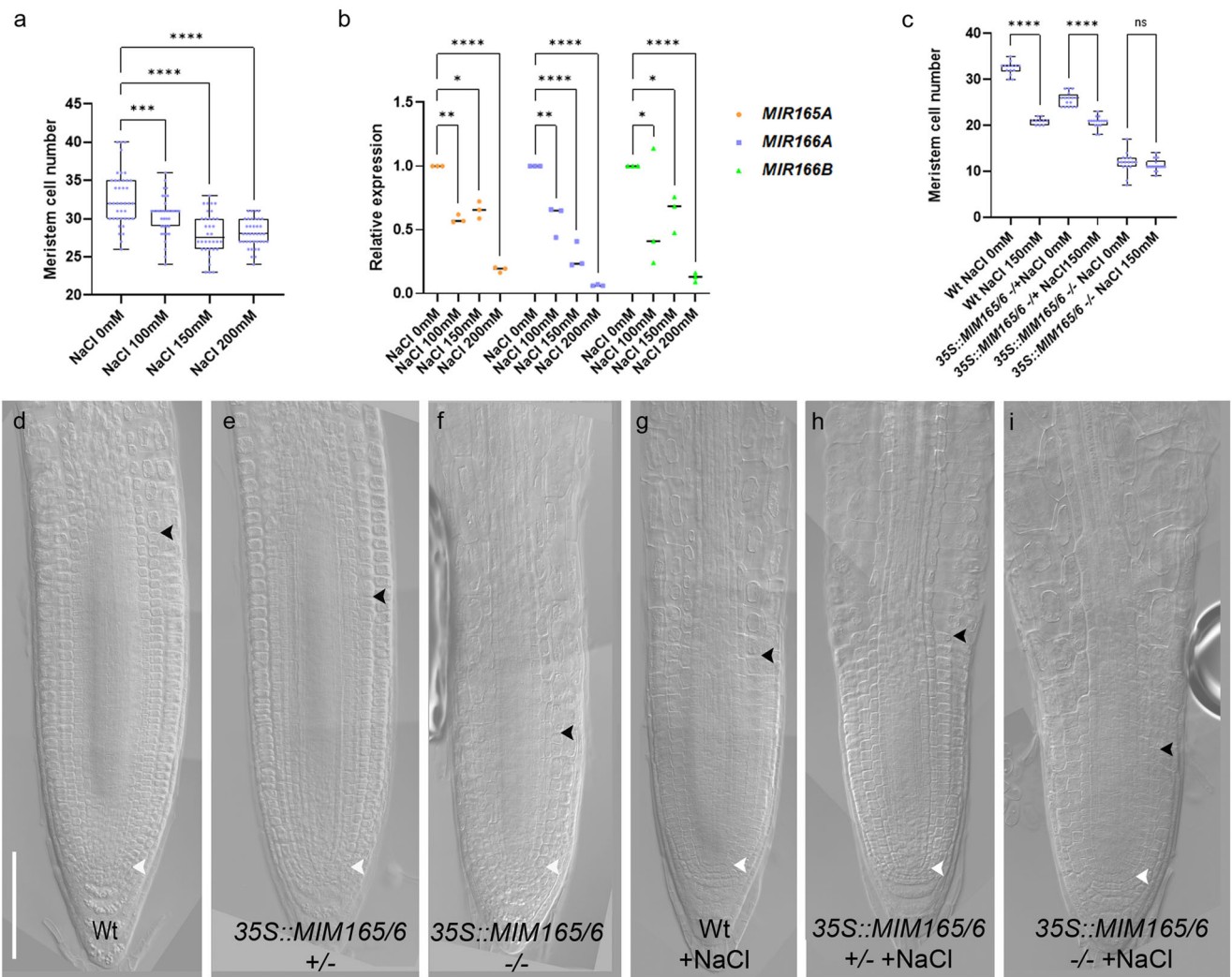

**Fig. 2 miR165 and 166 act in a dose dependent manner. a** Root meristem cell number of 5 dpg Wt root meristems exposed to 0, 100, 150, and 200 mM NaCl for 5 h (***$p = 0.0010$, ****$p < 0.0001$; One way ANOVA with post hoc Dunnett's multiple comparison test; $n = 40, 36, 34, 38$). **b** qRT–PCR analysis of *preMIR165A, preMIR166A, preMIR166B* RNA levels in the root tip of Wt plants upon 0, 100, 150, and 200 mM NaCl for 5 h (*$p < 0.0224$, **$p = 0.0078$, ****$p < 0.0001$; two-way ANOVA with post hoc Dunnett's multiple comparisons test; $n = 3$). **c** Root meristem cell number of 5 dpg Wt, *35S::MIM165/6* +/− (het), *35S::MIM165/6* −/− (homo) root meristems exposed to 150 mM NaCl for 5 h. (ns not significant, ****$p < 0.0001$ One way ANOVA with post hoc Sidak's multiple comparison test; $n = 14, 10, 16, 16, 16, 18$). DIC of 5 dpg root meristems of Wt (**d**), *35S::MIM165/6* +/− (**e**), *35S::MIM165/6* −/− (**f**) and of Wt (**g**), *35S::MIM165/6* +/− (**h**), *35S::MIM165/6* −/− (**i**) root meristems exposed to 150 mM NaCl for 5 h. Scale Bar 100 μm, white arrowheads indicate the cortical stem cell, black arrowheads the TZ. Box and whiskers plots show the median, 25th and 75th percentile (box limits), the 10th and 90th percentiles (whiskers), and outliers points.

specifically through the control of MIR165 and 166 expressions to adjust PHB levels, *phb-1d* plants should be resistant to salt exposure. As expected *phb-1d* plants were resistant to salt treatment, as their root meristem size did not vary upon NaCl exposure (Fig. 1o–s). These data suggest that NaCl specifically acts to decrease miR165/166 levels. Reduced miR165/166 levels result in an increase of PHB thus inducing root meristem arrest.

**The meristem dose–response to salt is regulated by miR165 and 166 levels**. Thus, we assessed whether salt had a dose-dependent effect on miR165 and 166 levels and, consequently, on root meristem size. First, we analyzed the response of the plant to different salt concentration. We found that lower concentrations (100 mM NaCl) affect root meristem size less than 150 mM and 200 mM (Fig. 2a). To assess whether this dose-dependent developmental effect was indeed due to a dose-dependent modulation of miR165 and 166 and PHB, we measured the levels of pre-

miR165A, 166A, 166B, and PHB in roots exposed for 2 h to 100, 150, and 200 mM NaCl, via qRT-PCR. Indeed, we observed a strict correlation between the increase of salt concentrations and the levels of miR165, 166, and PHB and the meristem size (Fig. 2b).

To further corroborate the causal relation between miR165/166 levels and salt response of the root meristem, we manipulated miR165 and 166 levels exploiting the mimicry technology, which employs molecules that sequester and destroy miRNAs thus diminishing free miRNA[35]. We generated plants that constitutively express a mimicry targeting both miR165 and miR166 (*35S::MIM165/166*). We analyzed the root meristem size of *35S::MIM165/6* lines in both homozygosity (*35S::MIM165/6*) and heterozygosity (*35S::MIM165/6* −/−) (Fig. 2c–f). Both homozygous and heterozygous plants showed a shorter root meristem than Wt, but *35S::MIM165/6* −/− homozygous plants displayed a shorter root meristem than the *35S::MIM165/6* −/− heterozygous plants (Fig. 2c–f). Moreover, *35S::MIM165/6* homozygous plants exposed to salt stress showed no root meristem size reduction,

possibly because miR165/166 levels in these roots were already too low to allow for further repression (Fig. 2c–i). On the other hand, NaCl treatment reduced root meristem size in 35S::MIM165/6 −/− heterozygous plants, although to a lesser extent than in Wt ones, presumably because of the salt-dependent downregulation of the residual miR165 and 166 levels (Fig. 2c–i). These data show that, upon salt stress, miR165 and 166 are responsible for the decrease in root meristem size in a salt-related dose-dependent manner.

Since miR165 and 166 modulate PHB levels, we set up to assess whether variations in PHB levels also affect root meristem size in a dose-dependent manner. We first analyzed the root phenotype of homozygous (phb-1d) and heterozygous (phb-1d/+) mutant plants. Both types of plants show shorter roots as compared to the Wt, with phb-1d roots being shorter than phb-1d/+ ones (Supplementary Fig. 3a–d).

The described phb-1d phenotype might in principle depend not only on a miR165 and 166 modulation, but also on a possible transcriptional regulation of PHB. Thus, we analyzed the root meristem of plants where the ectopic transcription of a miRNA-insensitive version of PHB, fused to the GREEN FLUORESCENT PROTEIN (phbmu-GFP), was driven by a UAS/GAL4 transactivation system, bypassing a putative regulation dependent on the PHB endogenous promoter. Among the several available transactivation lines, the Q0990 line was chosen because it drives expression only in the vascular tissue, where PHB is active. Q0990 ≫ phbmu-GFP plants show a shorter root meristem than Wt. Notably, in analogy with phb-1d mutant analysis, Q0990 ≫ phbmu-GFP homozygous lines display a more severe phenotype than heterozygous ones (Supplementary Fig. 3e–h). These data confirm the hypothesis that PHB-miR165/166 module is sufficient to cause variations in meristem size.

**Cytokinin mediates PHB response to salt exposure.** We have previously showed that PHB regulates meristem size activating the transcription of the cytokinin biosynthesis IPT7 gene[24]. Corroborating this, loss of IPT7 gene function in phb1-d/+ background shows a partial rescue of the phb-1d/+ root meristem phenotype (Fig. 3a–i). We therefore hypothesized that, in response to salt stress, PHB might modulate cytokinin levels, and hence root meristem size, via the regulation of IPT7 expression. To assess this, we first treated ipt7 loss-of-function mutant and phb-1d/+ ipt7 double mutant plants with 150 mM NaCl for 5 h. We found that salt treatment does not affect the size of ipt7 and phb-1d/+ ipt7 root meristems (Fig. 3a–i). This suggests that IPT7 is necessary to promote cytokinin-dependent cell differentiation in response to salt treatment. Then, we analyzed the expression of IPT7 in phb-1d and phb1-d/+, utilizing a nuclear-localized fluorescent transcriptional reporter of IPT7 (IPT7::3xGFP) (Fig. 3j–m). We observed that IPT7 expression expands to the vasculature and endodermis in phb-1d and phb1-d/+ compared with Wt, where is mostly expressed in the columella and lateral root cap (Fig. 3j–l). Moreover, a significantly higher GFP signal is detectable in the phb-1d homozygous line respect to the heterozygous one, thus suggesting that increased PHB levels are responsible for the enhanced IPT7 expression. Knowing that cytokinin promotes cell differentiation via the AHK3/ARR1/12 pathway, inducing the expression of the IAA3/SHORT HYPO-COTYL2 (SHY2) gene at the TZ[32,36], we assessed a possible involvement of this circuit in salt stress response. Thus, we analyzed the expression of SHY2 in roots after 2 h of NaCl exposure and found an induction of its expression (Supplementary Fig. 4). To assess whether salt-dependent regulation of PHB and PHV modulates SHY2 expression at the TZ, we treated phb phv SHY2::GUS plants with 150 mM NaCl. Interestingly, we

were unable to detect any SHY2 expression at the TZ in this background, neither before, nor after salt treatment (Supplementary Fig. 4), suggesting that the salt-dependent promotion of SHY2 expression depends on PHB.

These data suggest that salt exposure, by promoting cytokinin biosynthesis, enhances cytokinin-dependent cell differentiation activity at the TZ, hence, inhibition of root growth.

**Salt-controlled miR165/166 levels are critical to maintain root meristem function in response to salt stress.** It was already reported that PHB is at the core of an incoherent loop, involving cytokinin and miR165/166: cytokinin directly inhibits PHB through ARR1, while activating it through the modulation of miR166 and miR165, to control root meristem development. This incoherent loop allows a rapid homeostatic regulation of PHB in response to fluctuations of cytokinin, by maintaining an optimal threshold level of PHB necessary for proper root development[24]. We thus questioned whether this circuit acting to keep the robust development of the root would maintain PHB homeostasis also in response to salt. To investigate how salt stress impacts on the behavior of this circuit, we developed a mathematical model where the interactions between parameters are based on in-vivo experimental results. The model aims to understand how salt, cytokinin, miR165/166 and PHB, the components of the loop, react to changing salt concentrations, without determining the exact biochemical parameters of the system. According to the steady-state solutions of the model, as salt concentration increases miR165 and miR166 decreases, whereas PHB and cytokinin levels increase (Fig. 4a). For a more direct comparison with the experimental evidence, we simulated the system over time to determine the time course of the response of the circuit components to a salt-induced perturbation (Fig. 4b, c). According to the model, miR166/165 levels rapidly decrease after salt treatment. This causes PHB level to rise, resulting in cytokinin over-production, and in turn causing a decrease in meristem size and an inhibition of root growth. To understand how the levels of cytokinin are influenced by miR165/166-dependent PHB expression, we mimicked the miR165/166-insensitive PHB mutant (phb-1d) by performing two different time simulations of the model that, given the same initial conditions, differ for the inhibitory or non-inhibitory action of miR165/166 on PHB (dPHBmiR=0 in phb-1d) (Fig. 4d). The model showed how the lack of PHB inhibition raises the levels of cytokinin, thus reflecting phb-1d smaller meristem phenotype. Moreover, the simulation of phb-1d mutant after salt exposure, validate the experimental observation that the mutant shows salt resistance (Fig. 4e). In conclusion, our in-silico results confirm that the salt-mediated changes in mir166/165 levels are responsible for driving root developmental plasticity by regulating cytokinin production through PHB.

## Discussion
Altogether, our data show that miR165 and 166 control plastic development modulating PHB expression. This regulation is fundamental to adjust initial phasis of root plastic development in response to environmental cues, such as salt exposure. In particular, salt stress decreases levels of miR165 and 166 resulting in an increase of PHB expression that induces a salt-dependent increase of cytokinin level via the induction of IPT7 expression. Higher cytokinin levels at the TZ activates the AHK3/ARR1/12 pathway, which promotes the expression of SHY2, a negative regulator of the root putative morphogen auxin, triggering cell differentiation and represses root meristem activity in response to salt (Fig. 5)[32].

High cytokinin activity represses both the expression of PHB and miR165 and 166[24]. As previously suggested[24], and supported by our model, we posit that this regulation might help to maintain PHB levels within defined ranges in response to salt stress.

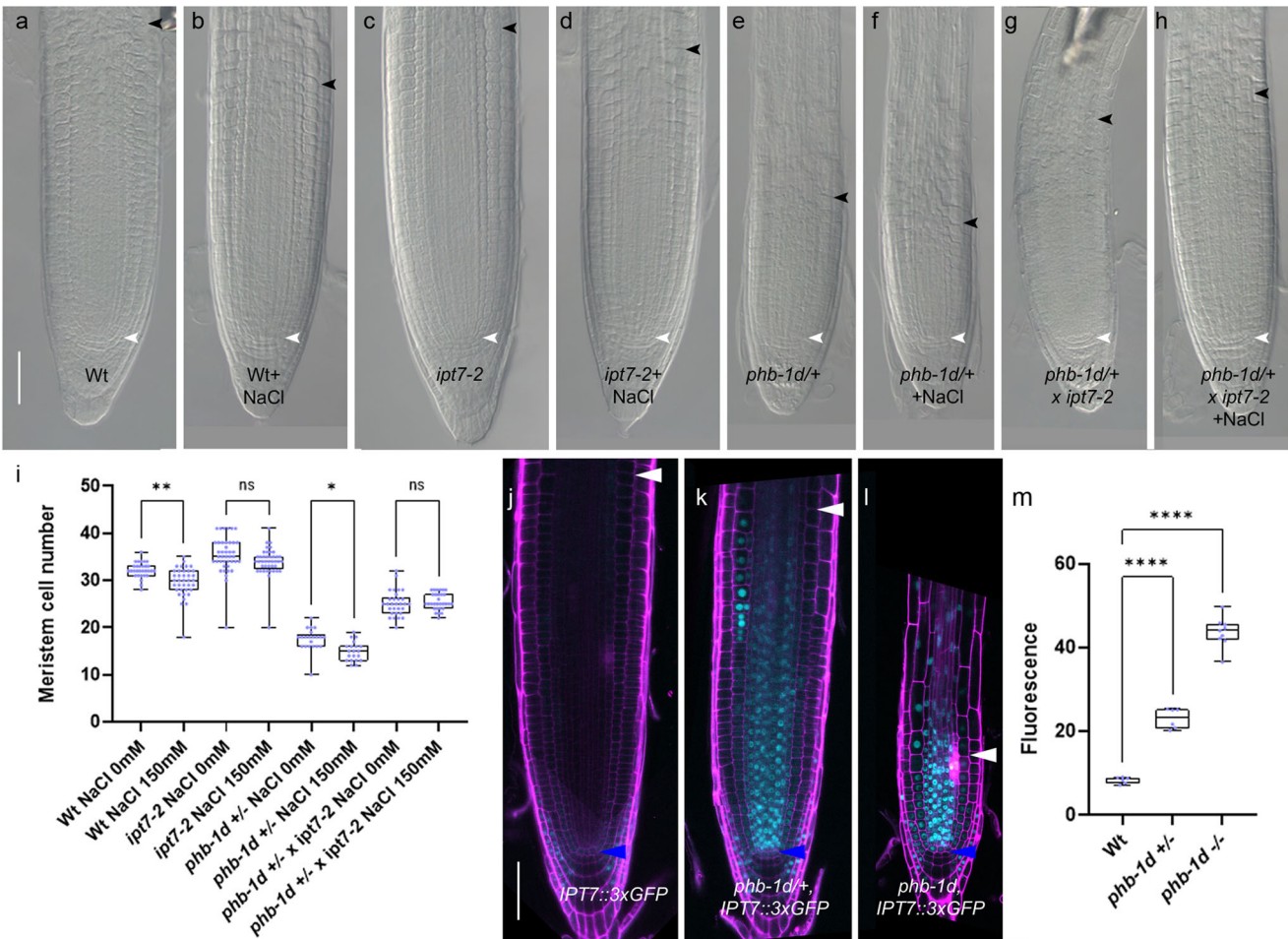

**Fig. 3 Salt stress inhibits root meristem activity via PHB/IPT7 module dosages.** DIC of 5 dpg root meristems of Wt (**a**), Wt exposed to 5 h 150 mM NaCl treatments (**b**), *ipt7-2* (**c**), *ipt7-2* exposed to 5 h 150 mM NaCl treatments (**d**), *phb-1d/+* (**e**), *phb-1d/+* exposed to 150 mM NaCl treatments for 5 h (**f**), *phb-1d/+ x ipt7-2* (**g**) and *phb-1d/+ x ipt7-2* exposed to 150 mM NaCl treatments for 5 h (**h**). Scale Bar 50 μm, white arrowheads indicate the cortical stem cell, black arrowhead the TZ. **i** Root meristem cell number of 5 dpg Wt, *ipt7-2*, *phb-1d/+*, *phb-1d/+ x ipt7-2* root meristems exposed to 0 mM and 150 mM NaCl for 5 h. (ns not significant, *$p = 0.0138$, **$p = 0.0011$; One way ANOVA with post hoc Sidak's multiple comparison test; $n = 30, 37, 41, 41, 21, 20, 30, 30$). Confocal images of 5 dpg root meristems of Wt (**j**), *phb-1d/+* (**k**) and *phb-1d* (**l**) carrying the construct *IPT7::3xGFP*. Scale Bar 50 μm, blue arrowheads indicate the cortical stem cell, white arrowheads the TZ. **m** Quantification of *IPT7::3xGFP* fluorescence in the vascular of the root meristem of WT, *phb-1d/+* and *phb-1d*. (****$p < 0.0001$; One way ANOVA with post hoc Dunnett's multiple comparison test; $n = 6, 6, 9$). Box and whiskers plots show the median, 25th and 75th percentile (box limits), the 10th and 90th percentiles (whiskers), and outliers points.

Hence, this incoherent loop might help to provide a fast recovery of root growth following salt stress exposure, maintaining robust root development. Our model serves as a valid approximation that reflects the experimental data, capturing the essential dynamics of the system, involving cytokinin, salt, and the miR165/166/PHB module. While acknowledging its simplicity, it provides a solid base for future research to build upon and investigate potential additional interactions. Despite the data provided are key for the regulation of plastic development, they do not fully explain how salt-dependent inhibition of root growth occurs during long salt exposure. Indeed, they only describe the initial phase of the stress response. Future integration of our data with computational models including tissue-specific gene activity in relation to growth will help gain a broader picture on the mechanism of action of the salt stress response.

Recently, it has been shown that genes involved in auxin catabolism, such as *GRATCHEN HAGEN 3.17* (*GH3.17*), are induced by salt stress[37]. *GH3.17* is a target of the AHK3/ARR1/12 module and, together with *SHY2*, generates an informative auxin minimum that triggers cell differentiation[38,39]. It will be interesting to assess in future whether salt-dependent regulation of

miR165 and 166 levels results in positioning the auxin minimum acting on *SHY2* and *GH3.17*.

Our results reveal that root adaptation to salt stress is initially driven by a modulation of miRNAs and miRNA target genes. Still to decipher is how and where salt is perceived in roots, and the details of the molecular pathways of salt stress response that leads to miR165 and 166 downregulation.

High-throughput single-cell RNA-sequencing assay[40–42] on roots exposed to different timing of salt stress might help to uncover the gene expression trajectories that alters cell transition in a high salt concentration context, but also to understand the trajectories that leads to a salt dependent repression of miR165 and 166 expressions. Recently, the gene expression trajectories that are altered by exogenous cytokinin applications have been identified[43]. It will be interesting combining the data obtained from single-cell RNA-sequencing assay on roots exposed to salt stress and exposed to cytokinin to uncover the spatial and time dependent molecular pathways through which salt stress inhibits root growth via this hormone.

Our data suggest a time-dependent inhibition of root meristem activity, as this happens only after few hours of high salt

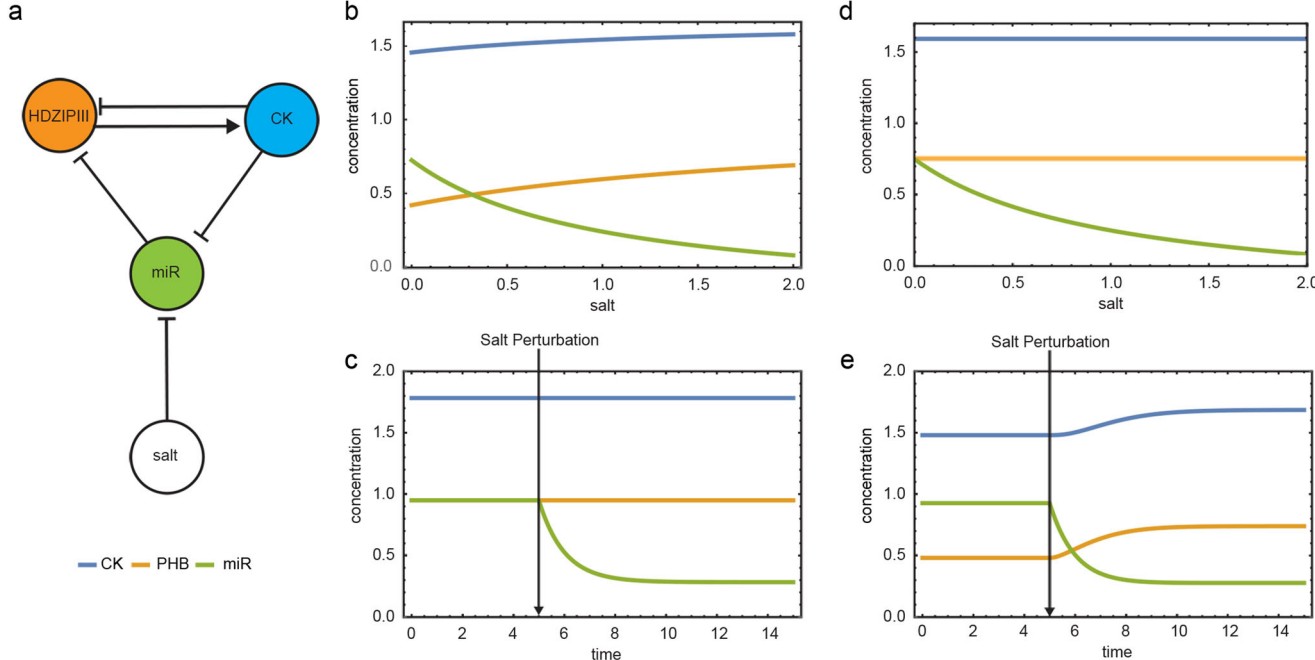

**Fig. 4 Salt-controlled miR165/166 levels are critical to determine root meristem size. a** Network topology of the model. Steady-State solutions of the model on a gradient of salt concentration in Wt (**b**), and in *phb1-d* (**d**). Time-course simulating salt perturbation from a steady-state in wild type (**c**), and in *phb1-d* (**e**).

treatment (Fig. 1). Based on our data, long exposures to high salt concentrations inhibit root growth independently of meristem activity. As cytokinin also drives cell elongation and the transition from elongating to differentiating cells[44], a possibility is that increased cytokinin levels in the root might promote root growth cessation, acting on these pathways. Also, high levels of cytokinin repress expression of *PHB* independently of miR165 and 166 regulation to maintain cytokinin homeostasis in the root[24]. Therefore, long exposition to high salt concentrations might alter different or additional molecular pathways.

It has been reported that salt stress induces abscisic acid (ABA) and ethylene production and signaling when root growth is inhibited, and it has been reported that ABA signaling promotes *MIR165A* expression to control vascular tissue plastic development[14,45]. In future works it will be interesting to understand how cytokinin, ABA and ethylene coordinate short- and long-term molecular dynamics that control root growth inhibition in response to salt stress.

Salt-dependent regulation of miR165/166 might not only induce the regulation of meristem activity, but also coordinate other strategies of salt stress adaptation. Plants adapt to salt also generating xylem gaps in the root, stabilizing the DELLA gibberellin repressors[46]. Salt-dependent reduction of miR165/166 levels might coordinate this adaptation strategy by promoting PHB and its direct target *GA2OX2*, an enzyme involved in gibberellin catabolism[30], thus reducing gibberellin levels and stabilizing DELLA proteins. The utilization of halophyte models, such as *Eutrema salsugineum*[47], could help to clarify whether and how alterations in miR165/166/PHB module promotes tolerance to salt stress.

## Materials and methods
**Plant material and stress treatments**. All mutants are in the *Arabidopsis thaliana* Columbia-0 (Col-0) ecotype background. For growth conditions, *Arabidopsis* seeds were surface sterilized, and seedlings were grown on one-half-strength Murashige and Skoog (MS) medium containing 0.8% agar at 22 °C in long-day conditions (16-h-light/8-h-dark cycle)[38]. Regarding stress treatments, 5 days old seedlings were grown on MS medium containing NaCl at different concentrations and for different periods of time as previously described in this paper.

*UAS::PHBmu-GFP, phb-13;phv-11*[24], *35S::MIM165/6* by ref. [35], *ipt7-2, phb-1d* and *phb,phv;SHY2::GUS* by ref. [24]. Enhancer trap line *Q0990* was obtained from the NASC. *Q0990»phbmu-GFP, phb1d;IPT7::3xGFP, phb-1d ipt7-2* were obtained by crossing. Homozygous and heterozygous lines were selected by phenotype.

**Generation and characterization of transgenic plants**. The *IPT7::3xGFP* plasmid was obtained as follow: 5.8 kb of *IPT7* promoter driving nuclear *3xGFP* were synthetized by GENEWIZ and inserted into *PMLBART* vectors via NotI flanking sites.

The *SCR::MIR166A* plasmid was obtained as follows: chpre-MIR166A was amplified from genomic DNA of Cardamine Oxford ecotype using specific primers: FW 5'-GGGGACAAGTTTGTACAAAAAAGCAGGCTGGGAGGAAGGAAGGGGCTTTCT-3' REV 5'-GGGGACCACTTTGTACAAGAAAGCTGGGTGCCCTAATTAAATTGAGAAGAAGG-3' and cloned in *pDONR221* Gateway vector by BP recombination (Invitrogen). *pDONRP4_P1-ChSCRp*[26] and pDONR221-chpre-miR166A were recombined with *pDONR P2R_P3-NOS* into a *pB7m34GW* destination vector via LR reaction (Invitrogen). *ChSCRp* has been utilized as it is expressed in the endodermis of *Arabidopsis thaliana* and it is not responsive to salt treatments (Supplementary Fig. 2a and b). Cardamine *pre-miR166A* (*chpre-MIR166A*) has been utilized to allow monitoring of endogenous Arabidopsis pre-miR166A response to salt as their sequences partially differ (Supplementary Fig. 2c,[26]). Cardamine miR166A and Arabidopsis miR166A mature forms are identical by sequence[26].

**Root length and meristem size analysis**. For root length measurements, plants were photographed and the resulting images were analyzed using the ImageJ 1.52t analysis software available online (https://imagej.nih.gov/ij/download.html).

Root meristem size for each plant was measured based on the number of cortex cells in a file extending from the quiescent center to the first elongated cortex cell excluded, as described previously[38]. The cortex is the most suitable tissue to count meristematic cells, as its single cell type composition shows a conserved number of cells among different roots. The boundary between dividing and differentiating cells for each tissue is called transition boundary (TB), while the region including the different transition boundaries is called transition zone (TZ)[31]. For root MC analysis, root meristems of 5 days post germination (dpg) plants were analyzed utilizing a differential Interference Contrast (DIC) with Nomarski technology microscopy (Zeiss Axio Imager A2). Plants were mounted in a chloral hydrate solution (8:3:1 mixture of chloral hydrate:water:glycerol) . Confocal images were obtained using a confocal laser scanning microscope (Zeiss LSM 780). For confocal laser scanning analysis, the cell wall was stained with 10 mM propidium iodide (Sigma-Aldrich). Student's t-test and ANOVA were used to determine statistical significance of these data (https://www.graphpad.com/quickcalcs/ttest1.cfm). Sample sizes are indicated for all experiments in the respective figure captions.

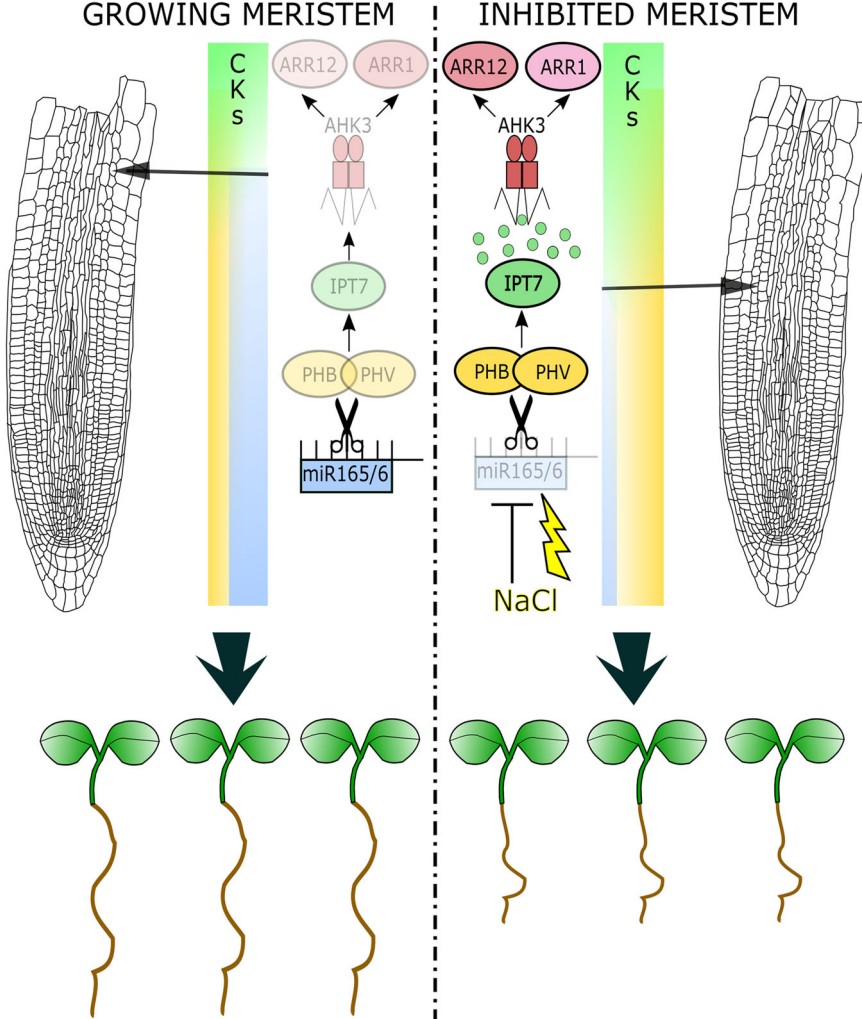

**Fig. 5 miR165/166 control root adaptation to salt stress.** Cartoon depicting root meristem and root seedlings growing in standard conditions (left) and exposed to salt stress (right). In standard conditions levels of miR165 and 166 (cyan bar) maintain low PHB/PHV levels (yellow bar) and hence low cytokinin (CKs) activity via IPT7/AHK3/ARR1/12 circuit. In salt stress conditions (NaCl, yellow bolt) decreased levels of miR165/166 (cyan bar) cause increased *PHB/PHV* levels (yellow bar) promoting cytokinin (CKs) activity via IPT7/AHK3/ARR1/12 circuit. This process regulates the differentiation rate (diff) and hence whole root length. See text for details. This figure was created using Inkscape.

**Table 1 qRT PCR primers.**

| Gene | Forward | Reverse |
|---|---|---|
| PHB | GCTAACAACCCAGCAGGACTCCT | TAAGCTCGATCGTCCCACCGTT |
| MIR165A | GATCGATTATCATGAGGGTTAAGC | CTATAATATCCTCGATCCAGACAAC |
| MIR166A | GGGGCTTTCTCTTTTGAGG | CGAAAGAGATCCAACATGAATAG |
| MIR166B | GATTTTTCTTTTGAGGGGACTGTTG | CTGAATGTATTCAAATGAGATTGTATTAG |
| UBQ10 | AATTGGAGGATGGTCGTACTTT | CAAAGTCTTGACGAAGATCTGC |

**RNA isolation and qRT-PCR**. Total RNA was extracted from roots of 5 days old seedlings (both controls and treated with NaCl) using the NucleoSpin RNA Plus (Macherey-Nagel). Reverse-transcription was performed using the Super-Script III First-Strand VILO cDNA Synthesis Kit (ThermoFisher Scientific). Quantitative RT-PCR (qRTPCR) analysis were performed using the gene-specific primers listed in Table 1.

All the primers used were tested for their qPCR efficiency of twofold amplifications per cycle by qRT-PCR with the Standard curve method. PCR amplifications were carried out using the SensiFast SYBR Lo-Rox (Bioline) mix and they were monitored in real time with a 7500 Real Time PCR System (Applied Biosystems). *UBIQUITINE 10* (*UB10*) amplification was used as housekeeper control and shown data are normalized to it. Data are expressed in $2^{-\Delta\Delta ct}$ value.

Three technical replicates of qRT-PCR were performed on at least two independent RNA batches. Results were comparable in all the experiments and with the housekeeper. Student's *t* test and ANOVA were performed to assess the significance of the differences between each sample and the control sample.

**Fluorescence image analysis**. GFP signal fluorescence of *IPT7::GFP* transgenic lines was acquired utilizing same pinhole and PMT parameters for each seedling/ image at 5 dpg. FIJI Image J software was used to quantify GFP signal intensity in the vascular tissue of the meristem of samples. The GFP intensity was reported as mean gray value. For each experiment, approximately six seedlings/images were examined, and three independent experiments were conducted.

**GUS histochemical assay**. β-Glucuronidase activity of transgenic lines carrying the GUS enzyme was assayed essentially as described in Moubayidin et al.12 using the β-glucoronidase substrate X-GlcA, (5-Bromo-4-chloro-3-indolyl-β-D-glu-curonic acid, Duchefa) dissolved in DMSO. X-GlcA solution: 100 mM Na₂HPO₄,

**Table 2 Model parameters.**

| Parameter | Details | Numerical Value |
|---|---|---|
| alphaCK | Baseline of cytokinin production | 1 |
| betaCK1 | Maximum rate of PHB induced CK synthesis | 1 |
| betaCK2 | Apparent dissociation constant for PHB regulation of CK synthesis | 0.5 |
| dCK | Rate of CK degradation | 1 |
| alphaPHB | Baseline rate of PHB transcription | 1 |
| betaPHB1 | Maximum rate of CK-inhibited PHB transcription | 1 |
| betaPHB2 | Dissociation constant for CK regulation of PHB transcription | 0.4 |
| dPHB | Baseline rate of PHB degradation | 1 |
| dPHBmiR | Rate constant for mir165/166-induced PHB degradation | 1 (0 in phb-1d mutant) |
| alphamiR | Baseline rate of mir165/166 transcription | 1 |
| betamiR1 | Maximum rate of CK-inhibited mir165/166 transcription | 1 |
| betamiR2 | Dissociation constant for CK regulation of mir165/166 transcription | 0.4 |
| dmiR | Rate of mir165/166 degradation | |
| salt | Contribution of the salt on the rate of basal miR165/166 transcription | 0 or 0.5 |
| n | Hill coefficient | 1 |

100 mM $NaH_2PO_4$, 0.5 mM K3 $K_3$ Fe(CN)$_6$, 0.5 mM $K_4Fe(CN)_6$, 0.1% Triton X-100 and 1 mg/ml X-GlcA. Seedlings were incubated at 37 °C in the dark for an appropriate time allowing tissue staining depending on the GUS line assayed. Imaging was done using the Axio Imager A2 (Zeiss) microscopy. Sample sizes are indicated for all experiments in the respective figure captions.

**Statistics and reproducibility**. Statistical analysis was performed using GraphPad (https://www.graphpad.com/). Student's t-test and ANOVA were used to determine statistical significance of these data (*$p < 0.05$, **$p < 0.01$, ***$p < 0.001$, ****$p < 0.0001$, ns not significant). All experiments have been performed in at least three replicas, sample sizes are indicated for all experiments in the respective figure captions. Box and whiskers plots show the median, 25th and 75th percentile (box limits), the 10th and 90th percentiles (whiskers), and outliers points.

**Model supplementary**. To describe our data, we devised a simple mathematical model of the biochemistry of salt-mediated cytokinin regulation. The model, like the one described in ref. [24], accounts for the time evolution of miR165, 166, PHB and cytokinin relative to salt concentration. The model consists of the following set of coupled, first-order, ordinary differential equations:

$$\begin{cases} \frac{\partial CK}{\partial t} = alphaCK + \frac{betaCK1 \cdot PHB^n}{bataCK2^n + PHB^n} - dCK \cdot CK \\ \frac{\partial PHB}{\partial t} = alphaPHB - \frac{betaPHB1}{1 \cdot \left(\frac{CK}{betaPHB2}\right)^n} - (dPHB \cdot dPHBmiR \cdot miR) \cdot PHB \\ \frac{\partial miR166}{\partial t} = \frac{alphamiR}{1 + salt} - \frac{betamiR1}{1 \cdot \left(\frac{CK}{betamiR2}\right)^n} - dmiR \cdot miR166 \end{cases} \quad (1)$$

where CK, PHB, and miR166 represent cytokinin, PHABULOSA and mir165/166 respectively. Adopting parameters as in ref. [24] (Table 2), we used the software Wolfram Mathematica to solve the equations in (1) for the steady-state with the free parameter *salt* in a range between 0 and 10 (Fig. 4). The system in (1) is solved numerically with parameters in Table using the Wolfram Mathematica NDSolve, which implements a Runge-Kutta method. To phenocopy the salt treatment of experiments on Arabidopsis plant, we use the value of the parameter salt equal to 0, and at time the parameter salt equal to 0.5 for a total time equal to 50 (Fig. 4).

**Reporting summary**. Further information on research design is available in the Nature Portfolio Reporting Summary linked to this article.

## Data availability

All data generated in this article are presented in Supplementary Data. Materials and data are available from the corresponding author upon request.

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

## Acknowledgements

We are grateful to Miltos Tsiantis for support through generation of materials and helpful discussions. To Patrizia Brunetti and Matteo Nava for thoughtful comments to the manuscript. This work was supported by a Giovanni Armenise-Harvard Foundation mid-career development grant (to S.S.), POR FESR Lazio 2014-2020 - code A0375E0189-2020-36640 (to G.B., N.S., S.S., R.D.I., P.V. and M.D.B.) and by Ministero dell'Istruzione, dell'Università e della Ricerca (MIUR). This research was funded by a seal of excellence grant from the Autonomous Province of Bozen/Bolzano (umor-D to S.J.U.).

## Author contributions

D.S., F.C., F.V., M.D.V., M.S., N.S., G.B., S.J.U R.D.M., P.V. performed experiments. E.S., M.T., conceived and generated mathematical models. D.S., M.D.B, R.D.M., S.S., R.D.I supervised experiments and analyzed data. R.D.I. conceived the research and designed the experiments. R.D.I. and P.C. wrote the paper. All authors commented on the manuscript.

## Competing interests

The authors declare no competing interests.
