## [Peer Review File · Communications Biology]

Reviewers' comments:

Reviewer #1 (Remarks to the Author):

This is a neat study that evidences the role of miR165 and 155 in shaping root response to salt stress. The topic is interesting, the experimental design is scientifically sound and the conclusion is solid. The finding will contribute to our understanding of the genomic mechanisms underlying developmental plasticity in plants. The manuscript was clearly written. To be acceptable, I suggest the authors to revise the manuscript in the following way.

(1) Although the authors presents an important topic in plant molecular biology, I feel that the manuscript lacks a global picture of how genes shape developmental plasticity. miR165 and 166 are important, but they may not be only determinants. A systematic view of genomic signature is needed. At least, the authors require a thorough discussion on this issue.

(2) The study used some mathematical equations. Yet, they are overly simple to describe a complex process. Equation 1 on the Model Supplementary page at line 486-487 works only when CK, PHB and miR166 co-exist in a condition that does not involve any other interactive agents. If there are any other interactive agents, parameters specifying this equation will change and, therefore, the result will change. I suggest that the authors interpret their results from this equation with caution.

(3) A technical issue: How many time points did you use to measure gene expression profiles for plants grown in salt stress and control? There are four measurement points at 5, 8, 16 and 24 hours after exposure to salt stress? To well fit equation 1, many more time points (say 10) are needed, in order to achieve a statistically confident result.

In summary, the manuscript should be acceptable after the above three issues are addressed or discussed.

Reviewer #2 (Remarks to the Author):

In this study, the authors discovered that microRNA165 and 166 are required for the regulation of root growth under salt stress. Salt stress inhibits the levels of miR165/166, which in turn leads to increased expression level of PHB. PHB is able to regulate the expression of cytokinin biosynthesis gene IPT7 and finally controls meristem size under high salinity. This study provides a novel insight into the molecular regulation of root elongation under salt stress, which will be of great interest to salt stress community.

1.Line82-83, the authors should explain why they selected miR165/166-PHB module to study the molecular basis of plants responding to salt stress.

2.Fig. 1K-1N and 1P-1S, whole seedlings grown on MS and MS+NaCl media can be presented.

3.Fig. 2b, PHB gene expression should also be analyzed in response to different concentrations of salt.

4.To genetically demonstrate that miR165/166, PHB, and IPT7 function in a pathway to regulate meristem size in response to salt stress, if the authors have related materials, it is suggested to test whether *phb phv* mutations suppress the phenotype of *35S::MIM165/6 -/-*, and whether *ipt7* mutation suppresses the phenotype of *phb-1d*.

Reviewer #3 (Remarks to the Author):

This study proved that salt stress promotes root meristem cells differentiation via reducing the dosage of the microRNAs miR165 and 166, and that miR165 and 166-dependent PHABULOSA (PHB)

modulation is fundamental for the response of root growth to this stress. The reductions of miR165 and 166 causes rapid increase of the PHB expression and the production of the root meristem pro-differentiation hormone cytokinin. The data provide direct evidence of how the miRNA-dependent modulation of transcription factors dosage mediates plastic development in plants. The paper had some new ideas and reference value.

However, the work presented in this manuscript contains several major flaws (see below) that must be addressed before it can be considered as a meaningful contribution.

Specific comments

1. The references are usually not cited in the Abstract section, please check the journal instruction or the paper published.
2. The introduction section should be divided into two or three paragraphs.
3. Line 37, signalling → signaling
4. Line 50, "microRNA molecules (miRNA) play a key role in the control of root meristem development" . here, the authors should give a few brief examples and show their roles.
5. Line 63, patterns → pattern
6. Line 86 and 88, line 89. "PHB" "miR165a, miR166a and 166b" should be in italic capitals type.
7. Fig1 G,H,I,J "PHB" "miR165a, miR166a and 166b" should be italic.
8. Line 96, MIR65A??
9. Line 110, the symbol before "These data" should be a period
10. Figure 4, mir165/166 → miR165/166
11. Figure 2 B, "preMIR165A" et al. There are different formats for "preMIR165/166" in the manuscript; please read through the MS to ensure the formats are uniform.

Our response in detail:

Reviewers' comments:

Reviewer #1 (Remarks to the Author):

This is a neat study that evidences the role of miR165 and 155 in shaping root response to salt stress. The topic is interesting, the experimental design is scientifically sound and the conclusion is solid. The finding will contribute to our understanding of the genomic mechanisms underlying developmental plasticity in plants. The manuscript was clearly written. To be acceptable, I suggest the authors to revise the manuscript in the following way.

(1) Although the authors present an important topic in plant molecular biology, I feel that the manuscript lacks a global picture of how genes shape developmental plasticity. miR165 and 166 are important, but they may not be only determinants. A systematic view of genomic signature is needed. At least, the authors require a thorough discussion on this issue.

We thank the reviewer for the suggestion, we elaborated on this point in the revised manuscript.

(2) The study used some mathematical equations. Yet, they are overly simple to describe a complex process. Equation 1 on the Model Supplementary page at line 486-487 works only when CK, PHB and miR166 co-exist in a condition that does not involve any other interactive agents. If there are any other interactive agents, parameters specifying this equation will change and, therefore, the result will change. I suggest that the authors interpret their results from this equation with caution.

We appreciate the reviewer's concern regarding the simplicity of the equations used in our study. However, our aim was to develop a model capturing the essential system dynamics, supported by robust experimental data involving cytokinins, salt, and the miR166/HD-ZIPIII module. We acknowledge that the presence of other interactive agents could potentially affect the parameters specifying the equations and alter the outcome. Nonetheless, our experimental data suggest that any additional interactions should work synergistically with the key components, as no conflicting results were observed.

(3) A technical issue: How many time points did you use to measure gene expression profiles for plants grown in salt stress and control? There are four measurement points at 5, 8, 16 and 24 hours after exposure to salt stress? To well fit equation 1, many more time points (say 10) are needed, in order to achieve a statistically confident result.

We thank the reviewer for the suggestion. Our study is focused on the mechanisms causing salt dependent inhibition of the root meristem activity that happens 4-5 hours after exposure to high salt concentrations (Fig.1). Hence, we focused mostly on the molecular mechanisms altered by salt stress in the first 4-5 hours from the treatment, and our model describes this timeframe. We agree with the reviewer that additional time points might be needed for fitting the equation, therefore we added 7 time points prior 5 hours in Figure 1. We also agree with the reviewer that additional information about later stages of root response to salt are needed and we have now elaborated on the discussion section different molecular pathways that might control entire root growth after 5 hours of salt stress treatment.

In summary, the manuscript should be acceptable after the above three issues are addressed or discussed.

Reviewer #2 (Remarks to the Author):

In this study, the authors discovered that microRNA165 and 166 are required for the regulation of root growth under salt stress. Salt stress inhibits the levels of miR165/166, which in turn leads to increased expression level of PHB. PHB is able to regulate the expression of cytokinin biosynthesis gene IPT7 and finally controls meristem size under high salinity. This study provides a novel insight into the molecular regulation of root elongation under salt stress, which will be of great interest to salt stress community.

1.Line82-83, the authors should explain why they selected miR165/166-PHB module to study the molecular basis of plants responding to salt stress.

We thank the reviewer. We added an explanation in the introduction section.

2.Fig. 1K-1N and 1P-1S, whole seedlings grown on MS and MS+NaCl media can be presented.

We thank the reviewer, however very small variations in root length happens in the Wt after 5 hours of salt treatments, therefore adding the picture of the whole seedling of Wt, phb-1d and SCR::MIR166 after 5 hours of salt treatment would not allow to appreciate developmental alterations due to the treatment.

3.Fig. 2b, PHB gene expression should also be analyzed in response to different concentrations of salt.

We thank the reviewer for this suggestion as this corroborates our results. We added this data in figure SD3.

4.To genetically demonstrate that miR165/166, PHB, and IPT7 function in a pathway to regulate meristem size in response to salt stress, if the authors have related materials, it is suggested to test whether phb phv mutations suppress the phenotype of 35S::MIM165/6 *-/-*, and whether ipt7 mutation suppresses the phenotype of phb-1d.

We thank the reviewer for this suggestion. We now included data regarding phb-1d/+,ipt7 double mutants exposed to salt stress. We decided to utilize phb-1d/+ as this strain is partially sensitive to salt stress. Indeed, homozygote phb1-d mutants are resistant to salt stress in a similar fashion to ipt7 mutants and no conclusion could be drawn exposing phb-1d,ipt7 double loss of function mutants to salt stress. As the reviewer can appreciate in figure 3, loss of ipt7 is sufficient to partially recover phb-1d/+ root meristem phenotype and to provide resistance to salt stress.

*Regarding testing whether phb phv mutations suppress the phenotype of 35S::MIM165/6 *-/-*, we agree with the reviewer that this experiment is interesting. However considering that MIM165/6 are specific to miR165/166 (Todesco et al., 2010, Yan et al., 2012, Jia et al., 2015), miR165/166 target only HD-ZIPIII (McConnell et al., 2001, Emmerly et al., 2003, Grigg et al., 2010, Carlsbecker et al., 2010, etch...), PHB and PHV are necessary and sufficient to regulate root development (Grigg et al., 2010, Dello Iorio et al., 2012, Sebastian et al, 2015, Di Ruocco et al., 2018) it is highly probable that phb phv mutations suppress the root phenotype of 35S::MIM165/6 *-/-*. Because this experiment is extremely time consuming and it would not add additional value to the focus of the study, we decided to not include it.*

Reviewer #3 (Remarks to the Author):

This study proved that salt stress promotes root meristem cells differentiation via reducing the dosage of the microRNAs miR165 and 166, and that miR165 and 166-dependent PHABULOSA (PHB) modulation is fundamental for the response of root growth to this stress. The reductions of miR165 and 166 causes rapid increase of the PHB expression and the production of the root meristem pro-differentiation hormone cytokinin. The data provide direct evidence of how the miRNA-dependent modulation of transcription factors dosage mediates plastic development in plants. The paper had some new ideas and reference value.

However, the work presented in this manuscript contains several major flaws (see below) that must be addressed before it can be considered as a meaningful contribution.

Specific comments

1. The references are usually not cited in the Abstract section, please check the journal instruction or the paper published.

We thank the reviewer for noticing this and we deleted citations from the abstract. Nonetheless, several articles published in Communications Biology report citations in the abstract.

2.The introduction section should be divided into two or three paragraphs.

3.Line 37, signalling→ signaling

4.Line 50, “microRNA molecules (miRNA) play a key role in the control of root meristem development”, here, the authors should give a few brief examples and show their roles.

5.Line 63, patterns → pattern

6.Line 86 and 88, line 89, “PHB” “miR165a, miR166a and 166b” should be in italic capitals type.

7.Fig1 G,H,I,J “PHB” “miR165a, miR166a and 166b” should be italic.

8.Line 96, MIR65A??

9.Line 110, the symbol before“These data”should be a period

10.Figure 4, mir165/166 →miR165/166

11.Figure 2 B, “preMIR165A”et al. There are different formats for "preMIR165/166" in the manuscript; please read through the MS to ensure the formats are uniform.

We thank the reviewer for noticing this major flaws, and we corrected through the text.

REVIEWERS' COMMENTS:

Reviewer #1 (Remarks to the Author):

The authors have satisfactorily addressed my comments.

Reviewer #2 (Remarks to the Author):

The authors have addressed most of my concerns.